rsos.royalsocietypublishing.org

power and energy systems/mechanical engineering

turbocharger, diesel engine, gas-path diagnosis, component health parameter, thermodynamic model

**Author for correspondence:**
Chuanlei Yang
e-mail: dalei2004@163.com

# A performance degradation evaluation method for a turbocharger in a diesel engine

Xinjie Cui[1], Chuanlei Yang[2], Jose Ramon Serrano[2] and Mingwei Shi[1]

[1]College of Power and Energy Engineering, Harbin Engineering University, Harbin 150001, People's Republic of China
[2]CMT-Motores Termicos, Universitat Politecnica de Valencia, Camino de Vera s/n, 46022 Valencia, Spain

 CY, 0000-0001-7468-5370

As one of the key systems of the marine power plant diesel engine, the turbocharger directly affects whether the diesel engine can continuously and stably provide the power required for the ship. Owing to a number of uncontrollable factors, such as harsh working conditions and complex structures, the turbocharger may have various failures, causing it to lose its intended function. At present, the fault diagnosis of the marine turbocharger has not been paid enough attention yet and in most cases, the method of 'ex post diagnosis' is still adopted. When analysing the nonlinear correspondence between the failure symptoms and failure causes, it is difficult for the existing theories to meet the actual diagnostic requirements. This paper introduces the concept of gas-path diagnosis into the condition monitoring for a marine turbocharger for the first time and proposes the flow capacity index which characterizes the flow capacity of the component and the isentropic efficiency index which characterizes the operating efficiency of the component as two dimensionless evaluation indicators for turbocharger health status. Moreover, the nonlinear mapping relationship between these two health parameters and the gas-path measurable parameters of the turbocharger is studied, and a novel performance degradation evaluation method for a turbocharger is established. The proposed method has been tested in three test cases where the degradation of a model turbocharger has been analysed. These case studies have illustrated that the proposed method can accurately isolate the degraded components and further quantify the degradation of the components.

# 1. Introduction

The marine exhaust turbocharger is mainly composed of a centrifugal compressor, a centripetal turbine, a rotating shaft, an oil and gas sealing element and a casing. As one of the key systems of a marine diesel engine, turbocharger plays an important role in ensuring the continuous and stable power supply of diesel engine and plays an important role in the impact of less exhaust emissions on the environment. With the increasing requirements for the dynamic performance and safety performance of diesel engine and the continuous development of science and technology, the structure of diesel engine and turbocharger is more and more complex and functions are more and more complete. As one of the key systems of the marine power plant diesel engine, the turbocharger directly affects whether the diesel engine can continuously and stably provide the power required for the ship to travel. Exhaust gas turbocharger has been widely used in a marine diesel engine. The marine exhaust turbocharger system is complex and its working environment is harsh, which makes it easy to fail during the operating process. Owing to a number of uncontrollable factors, such as harsh working conditions and complex structures, the turbocharger may have various failures, causing it to lose its intended function. In order to avoid serious accidents, it is necessary to use fault diagnosis technology to track its working status and conduct a health assessment. At present, the fault diagnosis of the marine turbocharger has not been paid enough attention yet, and in most cases, the method of 'ex post diagnosis' is still adopted. When analysing the nonlinear correspondence between the failure symptoms and failure causes, it is difficult the existing theory to meet the actual diagnostic requirements.

Through the investigation of the marine turbocharger-related enterprises and research institutions, the following problems exist in the fault diagnosis of marine turbocharger at present:

(i) In most cases, engineers rarely perform separate troubleshooting for marine turbochargers. Considering the relatively mature development of diesel engine technology, once the ship's power system fails, the maintenance personnel will first consider the failure analysis of the diesel engine. The turbocharger is rarely disassembled for troubleshooting, and the performance evaluation and fault diagnosis of the individual turbocharger has not received sufficient attention.

(ii) The current turbocharger fault diagnosis method mostly adopts single-machine and offline mode and has certain hysteresis. Various performance parameters monitored during the operation of the equipment are processed and analysed afterwards, to determine whether the fault has occurred and the fault type and its cause. This method is a typical 'ex post diagnosis' mode. It cannot reduce or prevent the occurrence of faults by performing effective manual intervention before the fault occurs.

(iii) The current automation and intelligence of marine turbocharger fault diagnosis are still very low. On the one hand, the current fault diagnosis method of marine turbocharger mainly relies on the experience of engineers and has strong subjectivity. For the complicated faults, it is difficult to locate and troubleshoot faults by manual diagnosis, and the diagnostic period is long and the diagnostic cost is high. In addition, the manual diagnosis method makes it difficult to predict the evolution trend of faults in the near future. On the other hand, the fault diagnosis process, including data collection, fault diagnosis and document data management, financial settlement, etc. is mostly done manually, which is inefficient and error-prone.

(iv) The fault diagnosis of marine turbocharger does not form a 'closed loop system' in its design and manufacturing. The fault diagnosis of the marine turbocharger is relatively independent, and there is a widespread phenomenon of 'not bad, not repaired, broken and repaired'. Few researchers have made deeper excavations of product failures, causing the same failures to occur repeatedly. It has also failed to improve product design and manufacturing processes, leading to a waste of data related to troubleshooting.

Therefore, it is necessary to adopt a new method based on the existing fault diagnosis theory and technology, to obtain a diagnostic model that can solve practical problems. The data-driven methods [1,2], such as pattern recognition [3–6] and machine learning [7], neural networks (NN) [8,9], Bayesian networks [10,11], fuzzy logic [12], support vector machine [13] and rough set theory [14], often need to be built on existing equipment fault sample sets. And these methods are often difficult to give accurate diagnostic results for fault types not covered in the sample. For a marine turbocharger, due to the lack of calibrated component failure sample data, it is difficult to establish a complete fault sample set that covers all fault types in a short period of time. And the accumulation of the relationship rule base for fault modes and fault symptoms through historical operational experience and on-site monitoring data is

rsos.royalsocietypublishing.org R. Soc. open sci. **5**: 181093

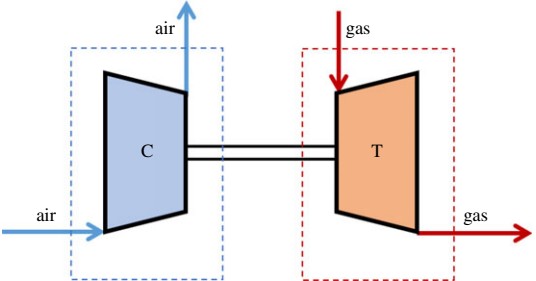

**Figure 1.** The thermodynamic system of the turbocharger.

a difficult and time-consuming task. And it is not easy to quantitatively evaluate the severity of the faults, which restricts the application of data-driven methods such as pattern recognition and machine learning. In the paper, a performance degradation evaluation method for turbocharger based on component-generalized characteristic map adaptation is proposed.

The remainder of this paper is organized as follows. In §2, the proposed performance degradation evaluation method is described. The description of the case studies is presented in §3, followed by the conclusion in §4.

# 2. Methodology

This paper introduces the concept of gas-path diagnosis into the condition monitoring of marine turbocharger for the first time and proposes the flow capacity index which characterizes the flow capacity of the component and the isentropic efficiency index which characterizes the operating efficiency of the component as two dimensionless evaluation indicators for the turbocharger health status. Moreover, the nonlinear mapping relationship between these two health parameters and the gas-path measurable parameters of the turbocharger is studied, and then a novel performance degradation evaluation method of the turbocharger is established.

## 2.1. Thermodynamic model of the turbocharger

In operation, the turbine end of the turbocharger is connected to the exhaust pipe of the diesel engine. The exhaust gas turbocharger drives the coaxial compressor impeller (about 30 000–120 000 r.p.m.) by using the exhaust gas energy (about 400–500°C and 0.2–0.4 MPa) discharged from the diesel engine, to achieve intake boost. The exhaust gas turbocharger and the diesel engine are not in mechanical contact with each other, and they transfer energy through an air flow or a gas flow, and the thermodynamic system of the turbocharger is shown in figure 1.

Both the compressor and the turbine are open systems. When the diesel engine is in stable operating condition, the flow of the working fluid (air or gas) can be regarded as a one-dimensional constant flow. The exhaust gas turbocharger meets the following three conditions during steady-state operation: (i) the energy between turbine and compressor is balanced; (ii) the turbine speed and compressor speed are equal; (iii) the flow of the working fluid through the turbine and compressor is balanced.

### 2.1.1. Thermodynamic model of the centrifugal compressor

The compressor of a marine diesel engine turbocharger is usually a single-stage centrifugal compressor. The characteristics of a compressor in the form of Cartesian coordinate's graph are often defined as compressor characteristic maps [15,16]. Compressor characteristic maps are used in the thermodynamic modelling for estimation of key component parameters, such as the pressure ratio $\pi_C$, the corrected mass flow rate $G_{C,cor}$ and the isentropic efficiency $\eta_C$ at several corrected rotational speeds $n_{C,cor}$. In practical application, $\pi_C$ and $\eta_C$ are expressed as a function of the $n_{C,cor}$ and $G_{C,cor}$, as shown in the equations (2.7) and (2.8), respectively.

According to similarity theory,

$$\pi_C = f_1(M_u, M_{ca}) = f_1\left(\frac{u}{\sqrt{kR_gT^*_{in}}}, \frac{c_a}{\sqrt{kR_gT^*_{in}}}\right) \tag{2.1}$$

and

$$\eta_C = f_2(M_u, M_{ca}) = f_2\left(\frac{u}{\sqrt{kR_gT_{in}^*}}, \frac{c_a}{\sqrt{kR_gT_{in}^*}}\right), \tag{2.2}$$

where $M_u$ is circumferential Mach number; $M_{ca}$ is axial Mach number.

Considering $P_{in}^*V = GR_gT_{in}^*$, the equations (2.1) and (2.2) can be converted into equations (2.3) and (2.4).

$$\pi_C = f_3\left(\frac{D \cdot n}{\sqrt{kR_gT_{in}^*}}, \frac{G\sqrt{R_gT_{in}^*}}{D^2P_{in}^*\sqrt{k}}\right) \tag{2.3}$$

and

$$\eta_C = f_4\left(\frac{D \cdot n}{\sqrt{kR_gT_{in}^*}}, \frac{G\sqrt{R_gT_{in}^*}}{D^2P_{in}^*\sqrt{k}}\right), \tag{2.4}$$

where $V$ is air volume flow rate at the inlet of the compressor; $D$ is the impeller diameter at the inlet of the compressor.

For the same compressor, the value of $D$ is constant and the equations (2.3) and (2.4) can be converted into equations (2.5) and (2.6).

$$\pi_C = f_5\left(\frac{n}{\sqrt{kR_gT_{in}^*}}, \frac{G\sqrt{R_gT_{in}^*}}{P_{in}^*\sqrt{k}}\right) \tag{2.5}$$

and

$$\eta_C = f_6\left(\frac{n}{\sqrt{kR_gT_{in}^*}}, \frac{G\sqrt{R_gT_{in}^*}}{P_{in}^*\sqrt{k}}\right). \tag{2.6}$$

As the change magnitude of the ratio of specific heat capacity $k$ is relatively small due to the varying ambient temperature, pressure and relative humidity, and the equations (2.5) and (2.6) can be converted into equations (2.7) and (2.8).

$$\pi_C = f_7\left(\frac{n}{\sqrt{R_gT_{in}^*}}, \frac{G\sqrt{R_gT_{in}^*}}{P_{in}^*}\right) = f_7\ (n_{C,cor}, G_{C,cor}) \tag{2.7}$$

and

$$\eta_C = f_8\left(\frac{n}{\sqrt{R_gT_{in}^*}}, \frac{G\sqrt{R_gT_{in}^*}}{P_{in}^*}\right) = f_8\ (n_{C,cor}, G_{C,cor}). \tag{2.8}$$

In practical application, the equations (2.7) and (2.8) can be further converted into generalized forms in equations (2.9) and (2.10), which represent compressor generalized nonlinear thermodynamic model.

$$G_{C,cor,rel} = f_9(n_{C,cor,rel}, \pi_{C,rel}) \tag{2.9}$$

and

$$\eta_{C,rel} = f_{10}(n_{C,cor,rel}, \pi_{C,rel}), \tag{2.10}$$

where $n_{C,cor,rel} = (n/\sqrt{T_{in}^* \cdot R_g})/(n_0/\sqrt{T_{in0}^* \cdot R_{g0}})$ is relative corrected spool-speed; $G_{C,cor,rel} = (G\sqrt{T_{in}^* \cdot R_g}/P_{in}^*)/(G_0\sqrt{T_{in0}^* \cdot R_{g\,0}}/P_{in0}^*)$ is relative corrected mass flow rate; $\pi_{C,rel} = \pi_C/\pi_{C0}$ is relative pressure ratio; $\eta_{C,rel} = \eta_C/\eta_{C0}$ is relative isentropic efficiency.

And the centrifugal compressor-generalized characteristic maps are shown in figure 2.

### 2.1.2. Thermodynamic model of the centripetal turbine

The turbine of a marine diesel engine turbocharger is usually a single-stage centripetal turbine. For the turbine-generalized characteristic maps, the deductive process is the same as the compressor-generalized characteristic maps, and the forms of the generalized relative corrected parameters are as follows, which represent turbine-generalized nonlinear thermodynamic model.

$$G_{T,cor,rel} = f_{11}(n_{T,cor,rel}, \pi_{T,rel}) \tag{2.11}$$

and

$$\eta_{T,rel} = f_{10}(n_{T,cor,rel}, \pi_{T,rel}), \tag{2.12}$$

where $n_{T,cor,rel} = (n/\sqrt{T_{in}^* \cdot R_g})/(n_0/\sqrt{T_{in0}^* \cdot R_{g0}})$ is relative corrected spool-speed; $G_{T,cor,rel} = (G\sqrt{T_{in}^* \cdot R_g}/P_{in}^*)/(G_0\sqrt{T_{in\,0}^* \cdot R_{g\,0}}/P_{in\,0}^*)$ is relative corrected mass flow rate; $\pi_{T,rel} = \pi_T/\pi_{T0}$ is relative pressure ratio; $\eta_{T,rel} = \eta_T/\eta_{T0}$ is relative isentropic efficiency (figure 3).

rsos.royalsocietypublishing.org   R. Soc. open sci. **5**: 181093

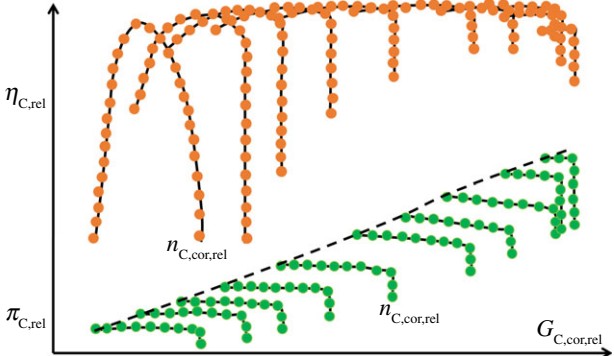

**Figure 2.** The centrifugal compressor generalized characteristic maps.

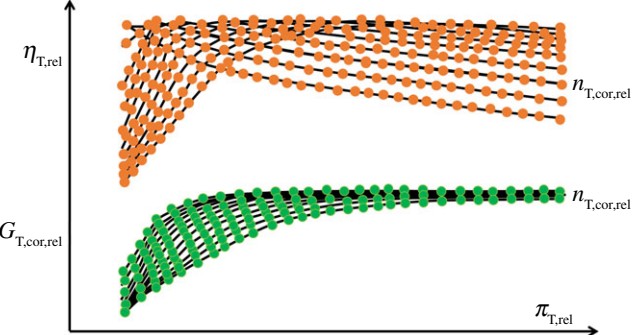

**Figure 3.** The centripetal turbine generalized characteristic maps.

**Table 1.** Dry air composition.

|        | volume fraction (%) | mass fraction (%) |
|--------|---------------------|-------------------|
| $N_2$  | 78.113              | 75.553            |
| $O_2$  | 20.938              | 23.133            |
| Ar     | 0.916               | 1.263             |
| $CO_2$ | 0.033               | 0.050             |

### 2.1.3. Thermophysical properties of air and gas

For the dry air, the components are usually fixed, as shown in table 1.

However, air usually contains water vapour, and the wet air composition needs to be calculated based on the current ambient temperature $T_0$, pressure $P_0$ and relative humidity $\phi$.

First calculate the humidity of the wet air:

$$\varphi = \frac{y_{H_2O}}{y_{air,dry}} = \frac{M_{H_2O}}{M_{air,dry}} \frac{\phi P_{H_2O,max}(T_0)}{P_0 - \phi P_{H_2O,max}(T_0)}, \tag{2.13}$$

where $y_{H_2O}$ is a mass fraction of the water vapour in the wet air; $y_{air,dry}$ is mass fraction of the dry air in the wet air; $\phi$ is the relative humidity; $P_{H_2O,max}(T_0)$ is the saturated water vapour pressure at ambient temperature $T_0$.

The mass fraction of the water vapour in the wet air can be determined from the humidity $\varphi$ of the wet air, and then the mass fraction of all components in the wet air can be obtained with the known mass fraction of each component in the dry air.

Compressed air at the outlet of the compressor enters the diesel engine and burns with the fuel $C_xH_yO_zN_uS_v$ to produce gas, and the combustion chemical reaction is shown in figure 4.

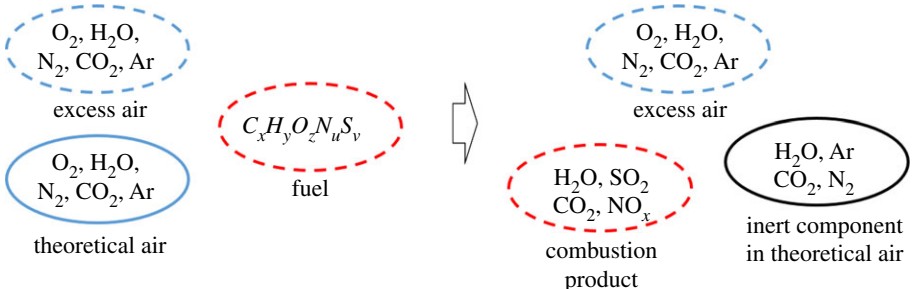

**Figure 4.** Combustion chemical reaction in the diesel engine.

The nitrogen element of the fuel $C_xH_yO_zN_uS_v$ usually generates $NO_x$ through combustion chemical reaction, but because of its extremely low content, it can be included in the final $N_2$ component during thermodynamic calculation. And the combustion chemical equation can be expressed as follows:

$$\beta C_xH_yO_zN_uS_v + \left(x + \frac{y}{4} + v - \frac{z}{2}\right)O_2 + d\left(x + \frac{y}{4} + v - \frac{z}{2}\right)N'_2 \rightarrow \beta\left(xCO_2 + \frac{y}{2}H_2O + vSO_2\right)$$
$$+ (1 - \beta)\left(x + \frac{y}{4} + v - \frac{z}{2}\right)O_2 + d\left(x + \frac{y}{4} + v - \frac{z}{2}\right)N'_2 + \frac{u}{2}\beta N_2. \tag{2.14}$$

According to the above combustion chemical reaction equation (2.14), the following conclusions can be obtained:

i. the theoretical consumption of wet air moles $n_{\beta=0}$ (i.e. the amount of wet air consumed when 1 mole of fuel is completely burned):

$$n_{\beta=0} = (1 + d)\left(x + \frac{y}{4} + v - \frac{z}{2}\right), \tag{2.15}$$

where $d$ is the volume ratio of nitrogen to oxygen in the wet air.

ii. theoretically generated the molar amount of gas $n_{\beta=1}$ (i.e. the molar amount of gas generated when 1 mole of fuel is completely burned):

$$n_{\beta=1} = n_{\beta=0} + \frac{y}{4} + \frac{z}{2} + \frac{u}{2}. \tag{2.16}$$

iii. The theoretical consumption of air quality $L_0$ (i.e. the mass of air consumed when 1 kg of fuel is completely burned):

$$L_0 = \frac{n_{\beta=0} \cdot M_{air(N_2+O_2)}}{M_{fuel}}. \tag{2.17}$$

where $M_{air(N_2+O_2)}$ is the molar mass of the air when only nitrogen and oxygen is taken into the air; $M_{fuel}$ is the molar mass of the fuel.

iv. The molar amount of gas $n_\beta$ generated when the fuel coefficient is $\beta$:

$$n_\beta = n_{\beta=0} + \beta\left(\frac{y}{4} + \frac{z}{2} + \frac{u}{2}\right). \tag{2.18}$$

v. The fuel coefficient $\beta$:

$$f = \frac{G_f}{G_a \cdot y_{O_2} + G_a \cdot y_{N_2}} \tag{2.19}$$

and

$$\beta = L_0 \cdot f, \tag{2.20}$$

where $G_f$ is the fuel mass flow into the diesel engine; $G_a$ is the air mass flow into the diesel engine; $y_{O_2}$ is the mass fraction of $O_2$ in the compressed air entering the diesel engine; $y_{N_2}$ is the mass fraction of $N_2$ in the compressed air entering the diesel engine.

rsos.royalsocietypublishing.org R. Soc. open sci. **5**: 181093

When the fuel coefficient is $\beta$, the molar fraction of each component in the gas can be obtained as follows:

$$
\left.
\begin{aligned}
r_{CO_2} &= x \cdot \frac{\beta}{n_\beta} \\
r_{H_2O} &= \frac{y}{2} \cdot \frac{\beta}{n_\beta} \\
r_{O_2} &= \left( x + \frac{y}{4} + v - \frac{z}{2} \right) \cdot \frac{(1-\beta)}{n_\beta} \\
r_{N'_2+N_2} &= \left[ d \cdot \left( x + \frac{y}{4} + v - \frac{z}{2} \right) + \frac{u}{2} \cdot \beta \right] \cdot \frac{1}{n_\beta} \\
r_{SO_2} &= v \cdot \frac{\beta}{n_\beta}.
\end{aligned}
\right\}
\tag{2.21}
$$

and

When the fuel coefficient is $\beta$, the molar mass of gas can be obtained as follows:

$$
M_{gas} = \sum_{i=1}^{5} M_i \cdot r_i,
\tag{2.22}
$$

where $M_{gas}$ is the molar mass of gas; $r$ is the molar fraction of each component in the gas; $M_i$ is the molar mass of each component in the gas.

The combustion gas component can be calculated from the known composition and mass of air and the known composition and mass of fuel by the above combustion chemical reaction equation. Taking into account the $H_2O$, $CO_2$, Ar in the excess air and in the theoretical air that are not involved in the combustion chemical reaction equation, the final actual gas composition can be obtained.

Through the above air component and gas component calculation process, the thermophysical properties of the current air and gas can be calculated according to the current working fluid temperature, based on the following ideal gas mixing formulae (2.23), (2.24) and (2.25) [17].

$$
M_{mixed} = \frac{m_{mixed}}{n_{mixed}} = \frac{\sum_{i=1}^{k} n_i \cdot M_i}{n_{mixed}} = \sum_{i=1}^{k} x_i \cdot M_i,
\tag{2.23}
$$

$$
c_{p,mixed} = \sum_{i=1}^{k} y_i \cdot c_{p,i}
\tag{2.24}
$$

and

$$
h_{mixed} = \sum_{i=1}^{k} y_i \cdot h_i,
\tag{2.25}
$$

where $M_{mixed}$ is the molar mass of air or gas; $y_i$ is the mass fraction of each component in air or gas; $c_{p,mixed}$ is the constant pressure specific heat capacity of air or gas; $h_{mixed}$ is the specific enthalpy of air or gas.

### 2.1.4. Thermodynamic model of the turbocharger

The mass conservation equation of the turbocharger can be obtained as follows:

$$
G_g = G_a + G_f.
\tag{2.26}
$$

The energy conservation equation of the turbocharger can be obtained as follows:

$$
N_C = N_T,
\tag{2.27}
$$

where $N_C$ is the compressor power consumption and $N_C = G_a(h_{out,C} - h_{in,C})/\eta_{m,C}$; $N_T$ is the turbine output power and $N_T = G_g(h_{in,T} - h_{out,T})\eta_{m,T}$.

## 2.2. Turbocharger health parameter definition

In the turbocharger operation, when some physical degraded problems of gas-path components happen, the component performance parameters $x$ (e.g. pressure ratio, mass flow rate and isentropic efficiency) are changing, and cause the deviation of gas-path measurable parameters $z$, such as temperatures, pressures and shaft rotational speeds, etc. Normally, turbocharger overall health status can be

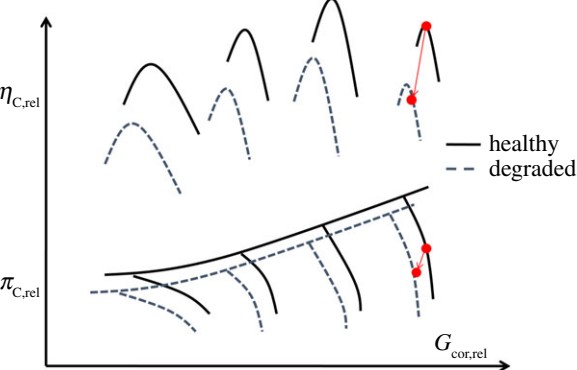

**Figure 5.** Compressor performance degradation or damage.

represented by gas-path component health parameters **SF** (i.e. compressor and turbine flow capacity indices and efficiency indices [18,19]), which represent a shift of the characteristic curves on component maps due to degradation, seen in figure 5. However, this essential performance and health status information cannot be directly measured and thus cannot be easily monitored and diagnosed.

### 2.2.1. Centrifugal compressor health parameters

$$SF_{\mathrm{C,FC}} = \frac{G_{\mathrm{C,cor,deg}}}{G_{\mathrm{C,cor}}}, \tag{2.28}$$

$$\Delta SF_{\mathrm{C,FC}} = \frac{G_{\mathrm{C,cor,deg}} - G_{\mathrm{C,cor}}}{G_{\mathrm{C,cor}}}, \tag{2.29}$$

$$SF_{\mathrm{C,Eff}} = \frac{\eta_{\mathrm{C,deg}}}{\eta_{\mathrm{C}}} \tag{2.30}$$

and
$$\Delta SF_{\mathrm{C,Eff}} = \frac{\eta_{\mathrm{C,deg}} - \eta_{\mathrm{C}}}{\eta_{\mathrm{C}}}, \tag{2.31}$$

where $SF_{\mathrm{C,FC}}$ is compressor flow capacity index; $G_{\mathrm{C,cor,deg}}$ is compressor corrected flow capacity when the compressor is degraded; $G_{\mathrm{C,cor}}$ is compressor corrected flow capacity when the compressor is healthy; $SF_{\mathrm{C,Eff}}$ is compressor isentropic efficiency index; $\eta_{\mathrm{C,deg}}$ is compressor isentropic efficiency when the compressor is degraded; $\eta_{\mathrm{C}}$ is compressor isentropic efficiency when the compressor is healthy.

### 2.2.2. Centripetal turbine health parameters

$$SF_{\mathrm{T,FC}} = \frac{G_{\mathrm{T,cor,deg}}}{G_{\mathrm{T,cor}}}, \tag{2.32}$$

$$\Delta SF_{\mathrm{T,FC}} = \frac{G_{\mathrm{T,cor,deg}} - G_{\mathrm{T,cor}}}{G_{\mathrm{T,cor}}}, \tag{2.33}$$

$$SF_{\mathrm{T,Eff}} = \frac{\eta_{\mathrm{T,deg}}}{\eta_{\mathrm{T}}} \tag{2.34}$$

and
$$\Delta SF_{\mathrm{T,Eff}} = \frac{\eta_{\mathrm{T,deg}} - \eta_{\mathrm{T}}}{\eta_{\mathrm{T}}}, \tag{2.35}$$

where $SF_{\mathrm{T,FC}}$ is turbine flow capacity index; $G_{\mathrm{T,cor,deg}}$ is turbine corrected flow capacity when the turbine is degraded; $G_{\mathrm{T,cor}}$ is turbine corrected flow capacity when the turbine is healthy; $SF_{\mathrm{T,Eff}}$ is turbine isentropic efficiency index; $\eta_{\mathrm{T,deg}}$ is turbine isentropic efficiency when the compressor is degraded; $\eta_{\mathrm{T}}$ is turbine isentropic efficiency when the compressor is healthy.

The effects of various types of gas-path faults on the flow capacity and operating efficiency of the components are shown in table 2.

**Table 2.** The effects of various types of gas-path faults on the flow capacity and operating efficiency of the components.

| gas-path faults | component flow capacity | component operating efficiency | category |
|---|---|---|---|
| compressor fouling | $SF_{C,FC}$ decrease | $SF_{C,EF}$ decrease | gradual |
| compressor erosion | $SF_{C,FC}$ decrease | $SF_{C,EF}$ decrease | gradual |
| compressor corrosion | $SF_{C,FC}$ decrease | $SF_{C,EF}$ decrease | gradual |
| compressor blade rubbing | $SF_{C,FC}$ decrease | $SF_{C,EF}$ decrease | gradual |
| turbine fouling | $SF_{T,FC}$ decrease | $SF_{T,EF}$ decrease | gradual |
| turbine erosion | $SF_{T,FC}$ increase | $SF_{T,EF}$ decrease | gradual |
| turbine corrosion | $SF_{T,FC}$ increase | $SF_{T,EF}$ decrease | gradual |
| turbine blade rubbing | $SF_{T,FC}$ increase | $SF_{T,EF}$ decrease | gradual |
| turbine thermal distortion | $SF_{T,FC}$ increase | $SF_{T,EF}$ decrease | gradual |
| object damage | $SF_{C,FC}$ decrease | $SF_{C,EF}$ decrease | abrupt |
| | $SF_{T,FC}$ decrease | $SF_{T,EF}$ decrease | |

## 2.3 Nonlinear mapping between health parameters and measurable parameters

The overall health status of a turbocharger normally is represented by component health parameters such as compressor and turbine flow capacity indices and efficiency indices, which virtually represent a shift of the characteristic curves on component characteristic maps due to degradations. However, these important performance and health status information cannot be directly measured and therefore are not easily monitored. During turbocharger operations, the deviation of component performance parameters can be indicated by the deviation of gas-path measurement parameters and such deviation of component performance parameters may be due to varying operating conditions or turbocharger performance degradation [18,19]. The thermodynamic relationship between turbocharger gas-path measurement parameters and turbocharger component performance parameters can be expressed with equation (2.36).

$$z = f(x, u) + v, \tag{2.36}$$

where $z$ is gas-path measurement parameter vector, $z \in R^M$; $x$ is turbocharger component performance parameter vector; $u$ is ambient and operating condition vector; $v$ is transducer measurement noise vector, $v \in R^M$.

Turbocharger gas-path fault diagnosis is an inverse mathematical problem to obtain the deviation of component performance parameters $\Delta x$ by the deviation of gas-path measurements $\Delta z$, and further to obtain the component health parameters $\Delta SF$ by comparing the current component performance parameters $x$ with those of initially healthy or clean turbocharger at component levels. Therefore, the thermodynamic relationship between component characteristic parameters and gas-path measurements can be further expressed as:

$$z = f(x, u) + v = f(map, \Delta SF, u) + v, \tag{2.37}$$

where $map$ is initially healthy or clean turbocharger component characteristic map vector; $\Delta SF$ is the gas-path component health parameter vector, $\Delta SF \in R^N$ and $N = 4$ for the turbocharger.

In order to uniquely determine these four component health parameters $\Delta SF$, the number and location of the measurable parameters for the turbocharger should be assigned reasonably, to ensure the quantity of the nonlinear equations, and the recommended turbocharger gas-path instrumentation set is as shown in table 3.

And then the performance evaluation method for a turbocharger can be shown in figures 6 and 7.

Here $E = z_{\text{deg}} - \hat{z}_{\text{deg}}$. Owing to the nonlinearity of turbocharger performance, an iterative process (a Newton–Raphson algorithm [16]) is used to obtain the component health parameters $\Delta SF$ based on component generalized characteristic maps adaptation until a converged solution is obtained when the predicted gas-path measurements $\hat{z}_{\text{deg}}$ are very close to the actual gas-path measurements $z_{\text{deg}}$ (i.e. the condition of $\|E\| < \varepsilon$ is satisfied and $\epsilon$ is a relatively small value).

rsos.royalsocietypublishing.org    R. Soc. open sci. 5: 181093

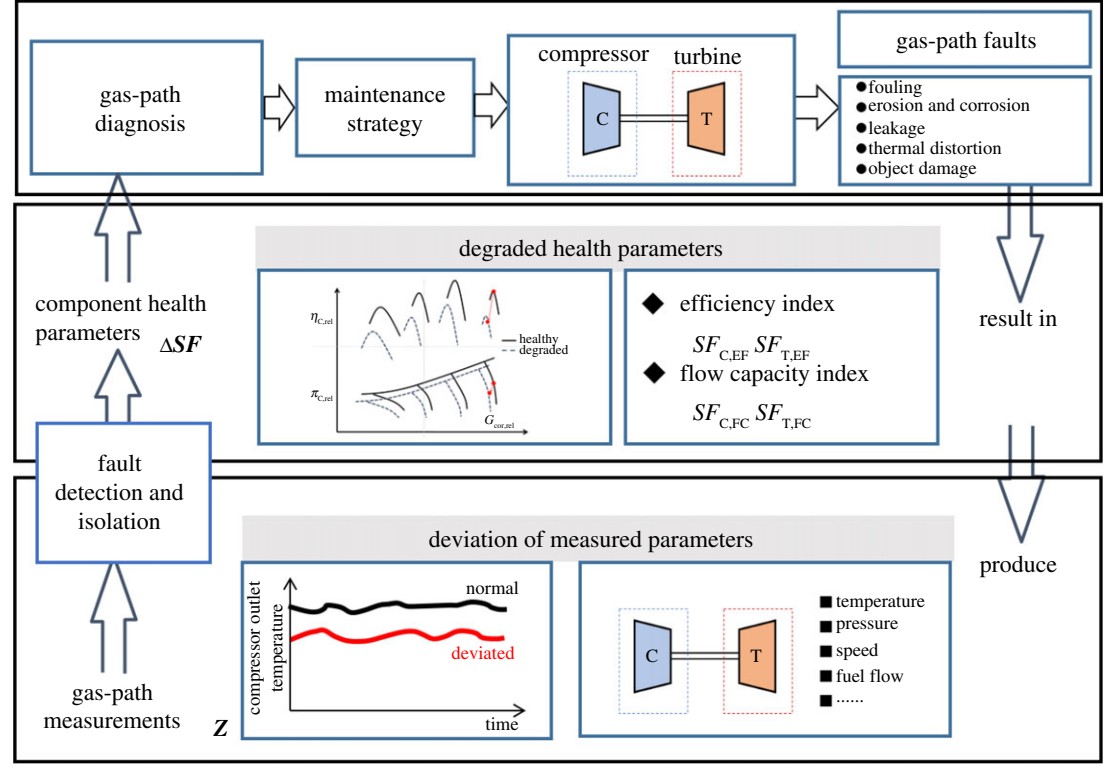

**Figure 6.** The schematic of the performance degradation evaluation method for turbocharger.

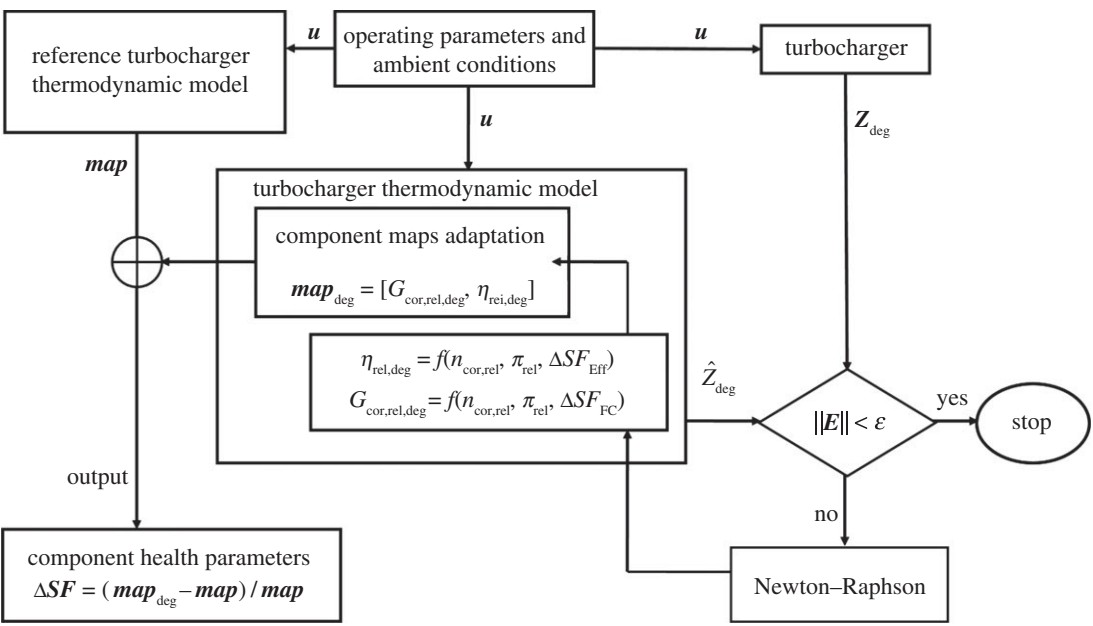

**Figure 7.** The performance evaluation method for turbocharger based on component generalized characteristic maps adaptation.

For the initial adaptation of the turbocharger thermodynamic model to the target turbocharger, the entire set of historical operating data is used for establishing a benchmark model that represents the clean/healthy condition of the target turbocharger, seen in figure 8. At the next stage, the objective of the diagnosis task is to deal with estimating the level of the component degradation of the target turbocharger. The performance adaptation is once again implemented for performing the diagnostic task. For a diagnostic purpose, the above tuning process is performed discretely for every new set of engine measurement data as seen in figure 8.

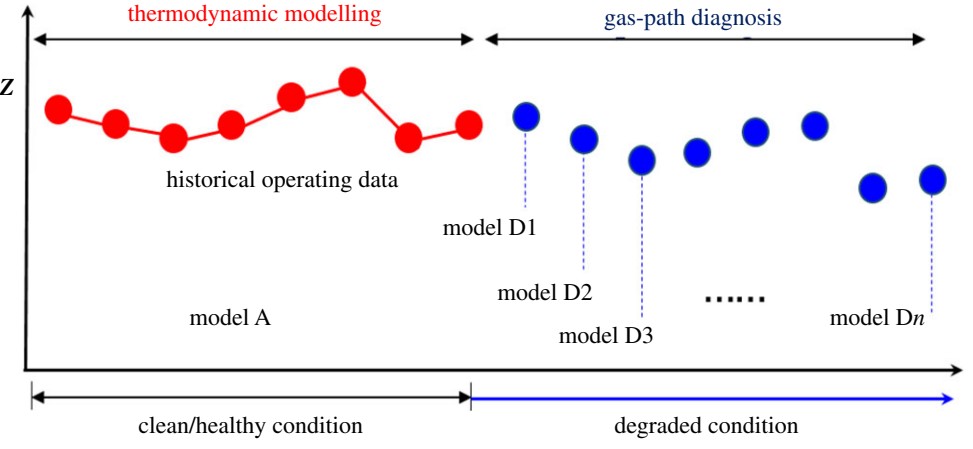

**Figure 8.** Representation of the turbocharger thermodynamic implemented for gas-path diagnosis.

**Table 3.** Turbocharger gas-path instrumentation set.

| parameter | compressor inlet pressure | compressor inlet temperature | fuel mass flow | compressor outlet temperature | compressor outlet pressure |
|---|---|---|---|---|---|
| symbol | $P_1$ | $T_1$ | $G_f$ | $T_2$ | $P_2$ |
| parameter | turbine inlet pressure | turbine inlet temperature | turbine outlet pressure | turbine outlet temperature | turbine speed |
| symbol | $P_3$ | $T_3$ | $P_4$ | $T_4$ | $n$ |

**Table 4.** Implanted major gas-path component degradations (%).

| component | centrifugal compressor | | centripetal turbine | |
|---|---|---|---|---|
| health parameter | $\Delta SF_{C,FC}$ | $\Delta SF_{C,Eff}$ | $\Delta SF_{T,FC}$ | $\Delta SF_{T,Eff}$ |
| mark no. | 1 | 2 | 3 | 4 |
| case 1 | $-2$ | $-2$ | 0 | 0 |
| case 2 | 0 | 0 | $+2$ | $-2$ |
| case 3 | $-2$ | $-2$ | $+2$ | $-2$ |

# 3. Result and analysis

The target turbocharger chosen for the demonstration of effectiveness of the proposed approach is a model turbocharger the same as the turbocharger performance model used in a marine diesel engine to bring convenience for testing the effectiveness of the proposed approach in quantity, seen in figure 1. The turbocharger thermodynamic model was created based on the simulation platform of Matlab software. The engine gas-path instrumentation set for performance evaluation of the model engine is shown in table 3.

To test the effectiveness of the approach, it is assumed that the compressor (C) and the turbine (T) of the model turbocharger may be degraded, and single or dual components may be degraded meanwhile. The degradation of the model turbocharger is simulated by changing the component health parameters $\Delta SF$ and three diagnostic cases shown in table 4 are used according to Diakunchak's experimental results [20].

To test the effectiveness of the proposed method, these sets of gas-path measurements (shown in table 3) simulated by implanting the various component degradations (shown in table 4) into the turbocharger performance model, respectively, are input to the proposed performance evaluation system described in §2,

rsos.royalsocietypublishing.org    R. Soc. open sci. 5: 181093

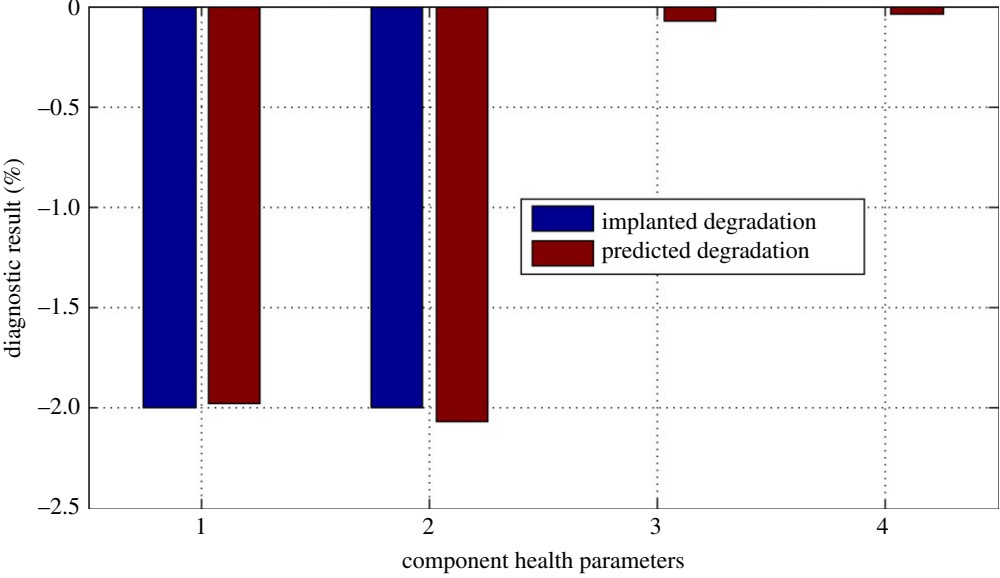

**Figure 9.** Diagnostic result for case 1.

**Table 5.** Maximum measurement noise.

| measurement | range | typical error |
|---|---|---|
| $P$ | 3∼45 psia | ± 0.5% |
| | 8∼460 psia | ± 0.5% or 0.125 psia |
| | | whichever is greater |
| $t$ | −65∼290°C | ± 3.3°C |
| | 290∼1000°C | $\pm \sqrt{2.5^2 + (0.0075 \cdot t)^2}$ |
| | 1000∼1300°C | $\pm \sqrt{3.5^2 + (0.0075 \cdot t)^2}$ |
| $G_f$ | up to 5450 kg h$^{-1}$ | 63.4 kg h$^{-1}$ |
| | up to 12 260 kg h$^{-1}$ | 142.7 kg h$^{-1}$ |

respectively, assuming that the degradations of the compressor and turbine are unknown. In this study, the simulated turbocharger performance with implanted component degradations is called 'actual performance' and the turbocharger performance predicted by using the proposed diagnostic system on the basis of gas-path measurable parameters is called 'predicted performance'.

As measurement noise is inevitable in actual gas-path measurements and can produce a negative effect on diagnosis, measurement noise is introduced in the simulated gas-path measurements to make the analysis more realistic. The maximum measurement noise for different gas-path measurements is according to the information provided by Dyson & Doel [21], as shown in table 5.

To reduce the negative effect of measurement noise on diagnostic analysis, multiple gas-path measurement samples are obtained in the simulation and a 30-point rolling average [17] was used to obtain an averaged measurement sample before the measurements are input to the proposed diagnostic system. The mathematical expression for the rolling averaging is shown in Equation (3.1).

$$\overline{z_i} = \frac{1}{P}\sum_{i=1}^{P} z_i, \tag{3.1}$$

where $z_i$ is $i_{\mathrm{th}}$ gas-path measurement samples and $P$ is the number of samples ($p = 30$ for 10-point rolling average).

Here cases 1 and 2 are used to test the effectiveness of the approach in isolating a degraded component and quantifying the degradation when only one component is degraded, and case 3 is used to test the effectiveness of the approach when dual components are degraded simultaneously.

rsos.royalsocietypublishing.org    R. Soc. open sci. **5**: 181093

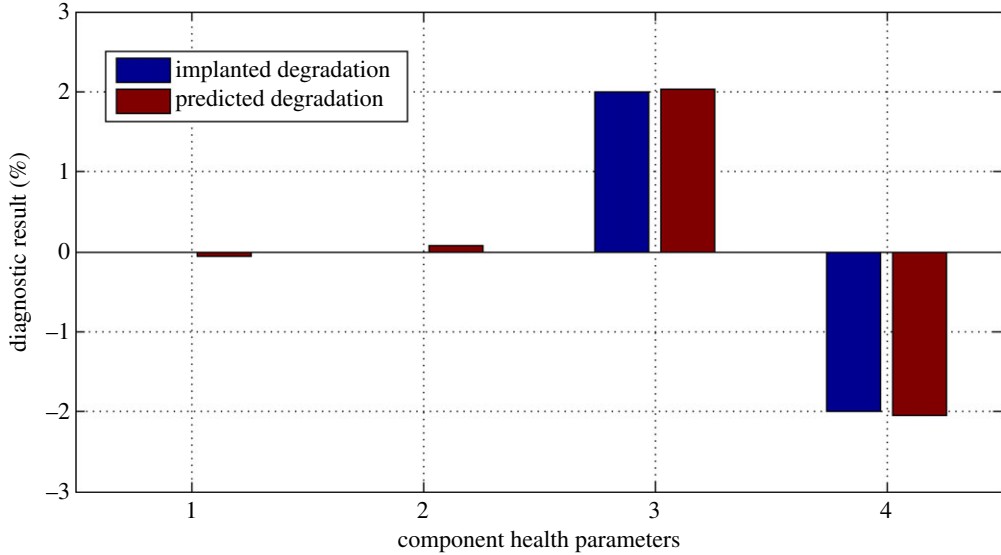

**Figure 10.** Diagnostic result for case 2.

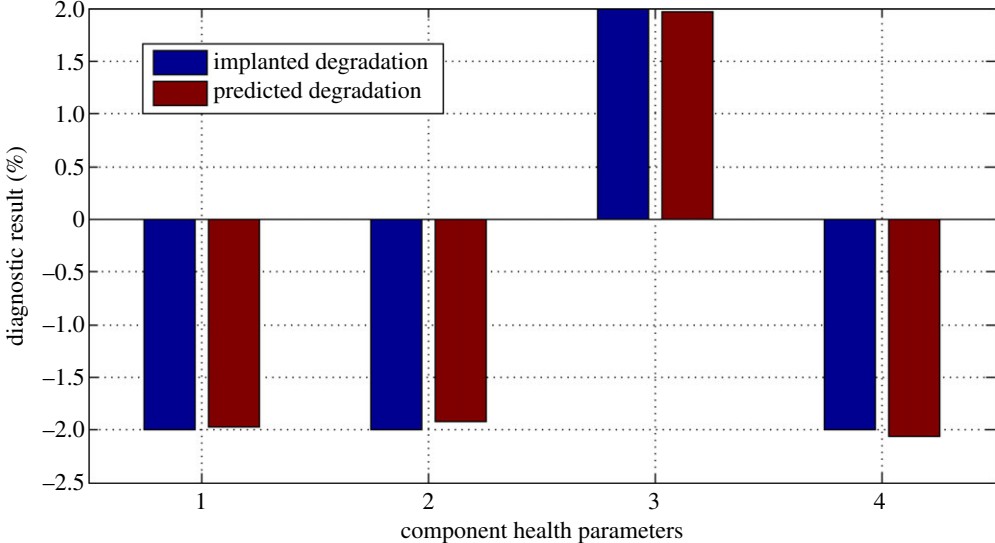

**Figure 11.** Diagnostic result for case 3.

And the diagnostic results are shown in figures 9–11.

From figures 9–11, it can be seen that due to the existence of gas-path measurement noise, the smearing effect can be found; however, the degraded component is successfully isolated and the magnitude of the fault degradation rate is obtained by the proposed performance evaluation method. The time cost of the proposed method over a laptop computer with a 4.0 GHz dual processor for one test case is 0.277 s.

## 4. Conclusion

In the paper, the concept of gas-path diagnosis is introduced into the condition monitoring of marine turbocharger for the first time, and two dimensionless evaluation indicators (i.e. the flow capacity index and the isentropic efficiency index) for the turbocharger health status are proposed as turbocharger health parameters. The nonlinear mapping relationship between these health parameters and the gas-path measurable parameters of the turbocharger is studied, and a performance evaluation method of the turbocharger is developed. Owing to the existence of gas-path measurement noise, the smearing effect can be found; however, the degraded component is successfully isolated and the

magnitude of the fault degradation rate is obtained by the proposed performance evaluation method. And the time cost of the proposed method is encouraging and the proposed method has the potential for real-time online monitoring.

Data accessibility. This article does not contain any additional data.

Authors' contributions. X.C. and C.Y. participated in data analysis, participated in the design of the study and drafted the manuscript; J.R.S. and M.S. proposed the performance evaluation method; all authors gave final approval for publication.

Competing interests. The authors declare that there is no conflict of interests regarding the publication of this paper.

Funding. The research of the paper is supported by the High-Tech Ship Research Project.

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
