## [Reviewer comments · Royal Society Open Science]

Review History

RSOS-181093.R0 (Original submission)

Review form: Reviewer 1

Is the manuscript scientifically sound in its present form?

Yes

Are the interpretations and conclusions justified by the results?

Yes

Is the language acceptable?

Yes

Is it clear how to access all supporting data?

Yes

Do you have any ethical concerns with this paper?

No

Have you any concerns about statistical analyses in this paper?

No

Recommendation?

Accept with minor revision (please list in comments)

Comments to the Author(s)

The authors proposed a performance degradation evaluation method for turbocharger in a diesel engine from the perspective of “gas-path diagnosis”, and the flow capacity index and the isentropic efficiency index as two dimensionless evaluation indicators for the turbocharger health status were firstly proposed. The case studies have illustrated that the method can accurately isolate the degraded components, and quantify the degradation for the components. The method proposed by the author has great potential application value for turbocharger diesel engine industry. The manuscript is well organized and is ready for publication with modification of written English throughout the paper to avoid grammatical errors.

Review form: Reviewer 2 (Georgios C. Mavropoulos)

Is the manuscript scientifically sound in its present form?

Yes

Are the interpretations and conclusions justified by the results?

Yes

Is the language acceptable?

Yes

Is it clear how to access all supporting data?

Yes

Do you have any ethical concerns with this paper?

No

Have you any concerns about statistical analyses in this paper?

No

Recommendation?

Accept as is

Comments to the Author(s)

The paper introduces for the first time the concept of gas-path diagnosis into the condition monitoring of marine turbocharger, and proposes two dimensionless evaluation indicators (i.e., the flow capacity index and the isentropic efficiency index) as turbocharger health parameters. The nonlinear mapping relationship between these health parameters and the gas-path measurable parameters of the turbocharger is studied, and a performance evaluation method of the turbocharger is developed. The proposed model is described and analyzed in detail and all the relevant equations are explained in clear inside the paper. The effectiveness of the proposed model is tested for three diagnostic cases for which experimental data are available in the literature. The results are very promising and reveal the

capacity of the proposed model to be used in real time condition monitoring since its necessary execution time is adequate for such applications.

The discussion and explanations provided by the authors about the findings are quite analytical and sufficient, stand in the appropriate level and do not extend into useless details. Quality of graphs is very good and the reader of the paper obtains easily and quickly an overview of the findings of the investigation performed. The subject of this investigation is quite important for the present status of technological development in the area of turbochargers and falls inside the scope of the journal. Therefore the paper is suggested for publication without any modification.

Dr.-Ing. Georgios Mavropoulos
Senior Research Associate
National Technical Univ. of Athens
Greece

Decision letter (RSOS-181093.R0)

27-Sep-2018

Dear Dr Yang,

The editors assigned to your paper ("A performance degradation evaluation method for turbocharger in a diesel engine") have now received comments from reviewers. We would like you to revise your paper in accordance with the referee and Associate Editor suggestions which can be found below (not including confidential reports to the Editor). Please note this decision does not guarantee eventual acceptance.

Please submit a copy of your revised paper before 20-Oct-2018. Please note that the revision deadline will expire at 00.00am on this date. If we do not hear from you within this time then it will be assumed that the paper has been withdrawn. In exceptional circumstances, extensions may be possible if agreed with the Editorial Office in advance. We do not allow multiple rounds of revision so we urge you to make every effort to fully address all of the comments at this stage. If deemed necessary by the Editors, your manuscript will be sent back to one or more of the original reviewers for assessment. If the original reviewers are not available, we may invite new reviewers.

- Data accessibility

<http://datadryad.org/submit?journalID=RSOS&manu=RSOS-181093>

- Competing interests

- Authors' contributions

- Acknowledgements

- Funding statement

Please note that Royal Society Open Science charge article processing charges for all new submissions that are accepted for publication. Charges will also apply to papers transferred to Royal Society Open Science from other Royal Society Publishing journals, as well as papers

submitted as part of our collaboration with the Royal Society of Chemistry (<http://rsos.royalsocietypublishing.org/chemistry>). If your manuscript is newly submitted and subsequently accepted for publication, you will be asked to pay the article processing charge, unless you request a waiver and this is approved by Royal Society Publishing. You can find out more about the charges at <http://rsos.royalsocietypublishing.org/page/charges>. Should you have any queries, please contact openscience@royalsociety.org.

on behalf of Prof. R. Kerry Rowe (Subject Editor)
openscience@royalsociety.org

Associate Editor's comments:

Please respond to the comments of the referees, and seek advice on the written English as recommended (<https://royalsociety.org/journals/authors/language-polishing/>).

Comments to Author:

Reviewers' Comments to Author:

Reviewer: 1

Comments to the Author(s)

The authors proposed a performance degradation evaluation method for turbocharger in a diesel engine from the perspective of “gas-path diagnosis”, and the flow capacity index and the isentropic efficiency index as two dimensionless evaluation indicators for the turbocharger health status were firstly proposed. The case studies have illustrated that the method can accurately isolate the degraded components, and quantify the degradation for the components. The method proposed by the author has great potential application value for turbocharger diesel engine industry. The manuscript is well organized and is ready for publication with modification of written English throughout the paper to avoid grammatical errors.

Reviewer: 2

Comments to the Author(s)

The paper introduces for the first time the concept of gas-path diagnosis into the condition monitoring of marine turbocharger, and proposes two dimensionless evaluation indicators (i.e., the flow capacity index and the isentropic efficiency index) as turbocharger health parameters. The nonlinear mapping relationship between these health parameters and the gas-path measurable parameters of the turbocharger is studied, and a performance evaluation method of the turbocharger is developed. The proposed model is described and analyzed in detail and all the relevant equations are explained in clear inside the paper.

The effectiveness of the proposed model is tested for three diagnostic cases for which experimental data are available in the literature. The results are very promising and reveal the capacity of the proposed model to be used in real time condition monitoring since its necessary execution time is adequate for such applications.

The discussion and explanations provided by the authors about the findings are quite analytical and sufficient, stand in the appropriate level and do not extend into useless details. Quality of graphs is very good and the reader of the paper obtains easily and quickly an overview of the findings of the investigation performed. The subject of this investigation is quite important for the present status of technological development in the area of turbochargers and falls inside the scope of the journal. Therefore the paper is suggested for publication without any modification.

Dr.-Ing. Georgios Mavropoulos
Senior Research Associate
National Technical Univ. of Athens
Greece

Author's Response to Decision Letter for (RSOS-181093.R0)

See Appendix A.

RSOS-181093.R1 (Revision)

Review form: Reviewer 1

Is the manuscript scientifically sound in its present form?

Yes

Are the interpretations and conclusions justified by the results?

Yes

Is the language acceptable?

Yes

Is it clear how to access all supporting data?

Yes

Do you have any ethical concerns with this paper?

No

Have you any concerns about statistical analyses in this paper?

No

Recommendation?

Accept as is

Comments to the Author(s)

The authors proposed a performance degradation evaluation method for turbocharger in a diesel engine from the perspective of "gas-path diagnosis", and the flow capacity index and the isentropic efficiency index as two dimensionless evaluation indicators for the turbocharger health

status were firstly proposed. The case studies have illustrated that the method can accurately isolate the degraded components, and quantify the degradation for the components. The method proposed by the author has great potential application value for turbocharger diesel engine industry. The manuscript is well organized and is ready for publication without any modification.

Review form: Reviewer 2 (Georgios C. Mavropoulos)

Is the manuscript scientifically sound in its present form?

Yes

Are the interpretations and conclusions justified by the results?

Yes

Is the language acceptable?

Yes

Is it clear how to access all supporting data?

Yes

Do you have any ethical concerns with this paper?

No

Have you any concerns about statistical analyses in this paper?

No

Recommendation?

Accept as is

Comments to the Author(s)

The paper in its present form is suggested for publication.

Decision letter (RSOS-181093.R1)

18-Oct-2018

Dear Dr Yang,

I am pleased to inform you that your manuscript entitled "A performance degradation evaluation method for turbocharger in a diesel engine" is now accepted for publication in Royal Society Open Science.

on behalf of Prof. R. Kerry Rowe (Subject Editor)
openscience@royalsociety.org

Associate Editor Comments to Author:
Congratulations on the acceptance of your paper.

Reviewer comments to Author:
Reviewer: 1

Comments to the Author(s)
The authors proposed a performance degradation evaluation method for turbocharger in a diesel engine from the perspective of “gas-path diagnosis”, and the flow capacity index and the isentropic efficiency index as two dimensionless evaluation indicators for the turbocharger health status were firstly proposed. The case studies have illustrated that the method can accurately isolate the degraded components, and quantify the degradation for the components. The method proposed by the author has great potential application value for turbocharger diesel engine industry. The manuscript is well organized and is ready for publication without any modification..

Reviewer: 2

Comments to the Author(s)
The paper in its present form is suggested for publication.

Appendix A

Journal title: Royal Society Open Science

Manuscript title: A performance degradation evaluation method for turbocharger in a diesel engine

Dear editor,

Thank you for your useful comments and suggestions on our manuscript. We have modified the manuscript accordingly, and the detailed corrections are listed below:

Reviewer(s)' Comments to Author:

Reviewer: 1

Comments to the Author(s)

The authors proposed a performance degradation evaluation method for turbocharger in a diesel engine from the perspective of “gas-path diagnosis”, and the flow capacity index and the isentropic efficiency index as two dimensionless evaluation indicators for the turbocharger health status were firstly proposed. The case studies have illustrated that the method can accurately isolate the degraded components, and quantify the degradation for the components. The method proposed by the author has great potential application value for turbocharger diesel engine industry. The manuscript is well organized and is ready for publication with modification of written English throughout the paper to avoid grammatical errors.

Answer: We have modified the written English throughout the paper, thanks for your advise!

Reviewer: 2

Comments to the Author(s)

The paper introduces for the first time the concept of gas-path diagnosis into the condition monitoring of marine turbocharger, and proposes two dimensionless evaluation indicators (i.e., the flow capacity index and the isentropic efficiency index) as turbocharger health parameters. The nonlinear mapping relationship between these health parameters and the gas-path measurable parameters of the turbocharger is studied, and a performance evaluation method of the turbocharger is developed. The proposed model is described and analyzed in detail and all the relevant equations are explained in clear inside the paper. The effectiveness of the proposed model is tested for three diagnostic cases for which experimental data are available in the literature. The results are very promising and reveal the capacity of the proposed model to be used in real time condition monitoring since its necessary execution time is adequate for such applications.

The discussion and explanations provided by the authors about the findings are quite analytical and sufficient, stand in the appropriate level and do not extend into useless details. Quality of graphs is very good and the reader of the paper obtains easily and quickly an overview of the findings of the investigation performed. The subject of this investigation is quite important for the present status of technological development in the area of turbochargers and falls inside the scope of the journal. Therefore the paper is suggested for publication without any modification.

Answer: thanks for your reviewing!